

# A Database of 10 min Average Measurements of Solar Radiation and Meteorological Variables in Ostrava, Czech Republic

Marie Opálková[1], Martin Navrátil[1], Vladimír Špunda[1], Philippe Blanc[2], Lucien Wald[2]

[1]Department of Physics, Faculty of Science, University of Ostrava, Ostrava, 70200, Czech Republic
[2]MINES ParisTech, PSL Research University, CS 10207 – 06904 Sophia Antipolis CEDEX, France

*Correspondence to*: Marie Opálková (opalkovamarie@seznam.cz)

**Abstract.** A database containing 10 min means of downward surface solar irradiance measured on a horizontal plane in several ultraviolet and visible bands from July 2014 to December 2016 at three stations in the area of the city of Ostrava (Czech Republic) is presented. The database contains time series of 10 min average irradiances or photosynthetic photon flux

densities measured in the following spectral bands: [280, 315] nm (UVB); [315, 380] nm (UVA); [400, 700] nm (photosynthetically active radiation, PAR); [510, 700] nm; [600, 700] nm; [610, 680] nm; [690, 780] nm; [400, 1100] nm. A series of meteorological variables including relative air humidity and air temperature at surface is also provided at the same 10 min time step at all three stations, and in addition air pressure, wind speed and wind direction at two stations. These two stations offer additional data: $PM_{10}$, $SO_2$, $NO_x$, $NO$, $NO_2$ concentrations. The details of the experimental sites and

instruments used for the measurements are given. Special attention is given to the data quality, and the process applied to label suspicious or erroneous measurements is described in detail. About 130 000 records for all three stations are available in the database. This database offers a unique ensemble of variables having a high temporal resolution and it is a reliable source on radiation in relation with environment and vegetation in highly polluted areas of industrial cities in the middle of Europe. The database has been placed on the PANGAEA repository (https://doi.pangaea.de/10.1594/PANGAEA.879722)

and contains individual data files of the three stations.

## 1 Introduction

Solar radiation is the crucial factor which influences the life on Earth. The intensity and spectral characteristics of incident solar radiation are important properties which cause physiological responses of plants, animals, and humans. Incident solar

radiation, especially in ultraviolet UVB [280, 315] nm, UVA [315, 400] nm and visible (VIS) [380, 760] nm spectral regions, is the key factor influencing plants. Although plants can adapt to changing environmental conditions, excessive radiation causes them acute damage because of over-excited photosystems and evolution of free radicals (Reddy and Raghavendra, 2006). Reversely, a lack of radiation is an important stress factor (Lambers et al., 2008). Plants use different spectral regions of solar radiation in different processes in their metabolism. Solar radiation can be a source of energy for





plant photosynthesis, mainly the blue and red components of photosynthetically active radiation (PAR, [400, 700] nm) (Ohashi-Kaneko et al., 2007; Johkan et al., 2010). It is also a source of information (UVB, UVA, blue and red components of PAR) as plants contain photoreceptors which are capable of recognizing individual spectral regions: UVR8 photoreceptor for UVB radiation (Tilbrook et al., 2013), cryptochromes and phototropins for UVA radiation (Verdaguer et al., 2017);

cryptochromes, phototropins and phytochromes for PAR (Casal, 2013). UVB is mostly harmful and plants try to avoid damage, by using, for example, UV shielding by producing chemical compounds which absorb UVB radiation (Meijkamp et al., 1999). UVA and the blue component of PAR are involved in phototropic reactions, in stomata activity, growth and plant development (Barillot et al., 2010). Plants are also significantly affected by the green light (Materová et al., 2017). Ratios of spectral regions of incident solar radiation, e.g. UVB/PAR or UVA/PAR, and ratio between diffuse and global radiation,

have also important effects on plants (Behn et al., 2010; Grifoni et al., 2008). Diffuse radiation spreads into the environment without a specific direction and is therefore able to penetrate within the plant canopy more effectively. Moreover, diffuse radiation is known for having different blue/red ratio compared with direct radiation (Navrátil et al., 2007), so diffuse radiation influences photosynthesis of plant canopies in different ways than the direct radiation. Although irradiance is lower in conditions when diffuse radiation is dominant, photosynthesis is more effective than under conditions of prevailing direct

radiation. Because air pollutants increase the diffuse fraction, industrial areas have a tendency to exhibit greater diffuse fractions than natural areas and interactions between radiation and plants will be different between these two areas. Since changes in both spectral composition and intensity of incident solar radiation can affect physiological processes in plants, detailed measurements of radiation in several spectral bands in UV and VIS ranges in highly polluted areas of industrial cities are useful for further understanding of these processes.

Solar radiation passing through the atmosphere is influenced by absorption and scattering due to gas, liquid and solid particles contained in the atmosphere (Wald, 2007). The character of cloudiness and smog periods in industrial urban areas cause considerable changes in the radiation environment compared with clean air areas. These changes can directly affect the photosynthetic activity and reduce the resistance of plants to other abiotic stresses. However, systematic measurements of radiation in UV and VIS spectral bands, which would allow finding and studying patterns of changes of intensity and

spectral quality of incident solar radiation in urban areas, are still missing. Several studies deal with a similar topic (e.g. Jacovides et al., 2004; Zhou & Savijärvi, 2014), but there is a lack of continuous measurements of irradiance in particular spectral regions of PAR, mainly blue ([400, 510] nm), green ([510, 600] nm) and red ([600, 700] nm) radiation, which are important for plants. The Ostrava industrial region experiences a great occurrence of smog periods (Jančík et al., 2013). The detailed measurements spanning several vegetative seasons performed at Ostrava aim at supporting a comprehensive

analysis of the changes in intensity and spectral composition of incoming irradiation in relation with air quality characteristics and other meteorological variables and further of the impact of air pollution on the radiation regime in that region and similar ones.



Because of existence of systematic errors in radiation measurements (e.g. Muneer and Fairooz, 2002; Younes et al., 2005), particular attention was devoted to a careful quality check (QC) on the measurements (e.g. Journée and Bertrand, 2011; Roesch et al., 2011) before releasing the data for further analyses.

The aim of this paper is to describe a database elaborated in cooperation between the University of Ostrava (Department of Physics, Faculty of Science) and the research centre Observation, Impacts, Energy of MINES ParisTech. The database covers the period 2014–2016 and includes quality-checked 10 min averages of seven spectral regions within UV and VIS as well as the broadband irradiance at three measuring stations situated at two locations within the wider area of the city of Ostrava. Meteorological variables, such as air temperature and relative air humidity, are also included in the database for all three measuring stations. Other meteorological variables, such as air pressure, wind direction, wind speed and concentrations of several air pollutants, are included for one location.

This paper is organized as follows: the study area and details about sensors for radiation measurements are given in Sect. 2, the QC procedures applied to the measurements are described in Sect. 3, their results and the structure of the database are detailed in Sect. 4. Finally, a summary and perspectives are given in Sect. 5. The quality-checked database is available in PANGAEA (https://doi.pangaea.de/10.1594/PANGAEA.879722) under Creative Commons Attribution 3.0 Unported International Public License (CC-BY 3.0).

## 2 Ground stations and measurements

### 2.1 Locations of the measurement sites

The city of Ostrava is located in the north-eastern part of the Czech Republic. Mean elevation of this area is 244 m a.s.l. and the countryside is flat or slightly undulating (Weissmannová et al., 2004). According to the Köppen-Geiger classification, this area belongs to the DFb climate subtype (Peel et al., 2007) – mild warm climate, in addition with urban influences. Summers are long, warm and mildly dry, winters are short, mildly warm with a short duration of snow cover (50–60 days). Mean annual duration of sunshine is 1 594 h, mean annual air temperature is 8.4 °C (with the greatest value in July: 18.3 °C), and mean annual precipitation is 700 mm (with the greatest value in July: 95.6 mm; Weissmannová et al., 2004).

Three identical systems of sensors were installed at two different locations. Two systems were installed in the Botanical Garden of the University of Ostrava (BG OU) in Ostrava-Radvanice (stations S1 and S2, Tab. 1, Fig. 1). The third one (station S3) was installed in the area of the Czech Hydrometeorological Institute (CHMI) in Ostrava-Poruba. Both locations are at the same altitude of ~ 240 m a.s.l. (Tab. 1, Fig. 1). BG OU is situated approximately 3 km from an industrial area which produces many air pollutants (Jančík et al., 2013) and is much more influenced by air pollution than the CHMI location, especially in the winter months. It was not possible to place the stations in a very open space area due to necessity of a fenced and good approachable area. There are some shrubs and trees in the surroundings of the sensors at BG OU and other measuring devices surrounding the CHMI sensors. Their shading effects were studied and included in the database. Shaded and non-shaded data were determined visually by coloured plots of the irradiance as a function of the solar azimuth



and elevation angles (figures with details are included in the datasets). These horizons were confirmed by the analysis of the digital elevation model found in Google maps.

**Tab. 1: List of measuring stations with some details. "BG OU" means Botanical Garden of University of Ostrava, "CHMI" means Czech Hydrometeorological Institute.**

| Station | Abb. | Location | District | GPS | Period of measurements |
|---------|------|----------|----------|-----|------------------------|
| Station 1 | S1 | BG OU | Radvanice | 49.82754°N, 18.32618°E | Jul 2013 – Dec 2016 |
| Station 2 | S2 | BG OU | Radvanice | 49.82751°N, 18.32615°E | Jul 2014 – Dec 2016 |
| Station 3 | S3 | CHMI | Poruba | 49.825212°N, 18.159319°E | Jul 2014 – Dec 2016 |

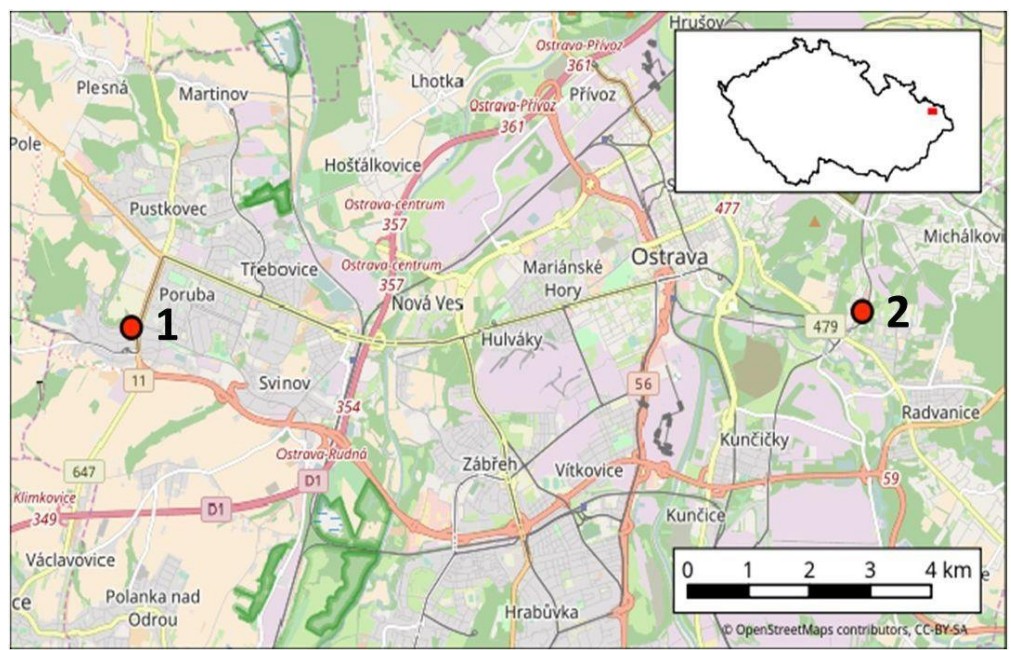

**Fig. 1: Locations of the measuring stations. 1: Czech Hydrometeorological Institute area (station S3); 2: Botanical Garden of the University of Ostrava (stations S1 and S2).**

### 2.2 Sensors, measurements, data storage, maintenance and calibration

Several types of sensors were installed to measure the downwelling solar radiation or the photosynthetic photon flux density (PPFD) received at the ground level on a horizontal plane in different spectral bands. The database contains measurements from July 2014 to December 2016. Data loggers collected signals from the sensors each minute. The raw data was stored as 10 min averages to ease further numerical processing.

At all three stations, there were sensors which measured global radiation in UVB, UVA, broadband, and the PPFD at various wavelength ranges within the PAR band: [400, 700] nm, [510, 700] nm, and [600, 700] nm, and spectral bands around 660 nm and 730 nm (Tab. 2). The PPFDs in three PAR bands were calculated from the sensor data: blue [400, 510] nm,



green [510, 600] nm, and red [600, 700] nm by subtraction. The data was collected by data loggers Edgebox V12 from the EMS Brno company (http://www.emsbrno.cz). At all three stations, air temperature and relative air humidity were measured (Tab. 2). Technical characteristics of the sensors are available at the website of the Skye company (http://www.skyeinstruments.com/ – for UVB, UVA, red and far red sensors), and at the website of the EMS Brno for all

other sensors except custom-made sensors for spectral intervals [510, 700] nm and [600, 700] nm, whose basis was the sensor EMS 12.

Each radiation sensor was equipped with cosine correction. Maintenance of sensors was performed every month in winter, when heavy loads of dust particles in the atmosphere are frequent, and every two months during the rest of the year. The sensors were cleaned with pure ethanol and the silica gel for removing air humidity within the data logger was changed at

each maintenance.

The sensors were calibrated before the start of the measurements in 2013 for the station S1 and in 2014 for the stations S2 and S3. EMS Brno recommends recalibration of their own sensors once per five years since all sensors are equipped with glass diffusers, which allow accurate measurements for a long time. To check the long-term stability of the sensors, measurements from each of them were compared with the measurements in the broadband range and linear regressions were

computed in the whole measuring period and for each individual year. No significant changes in the values of the regression parameters were observed. In addition, at BG OU, measurements made by similar sensors were compared to check the temporal consistency as the stations were only 3 m apart. Coefficients of determination were in the interval [0.94, 0.98], thus confirming the expected similarity in data between S1 and S2. No significant systematic biases were observed and the least square linear regression provided a cross-calibration correction of less than 5%. Differences between them could be caused

by unequal irradiance during partly cloudy days, or by technical properties of sensors. For example, sensors measuring radiation in the intervals [510, 700] nm and [600, 700] nm contain cut-off filters which have the S-shaped permeability curve and it causes a little bit different measured values. Based on these tests, we can conclude that no long-term decrease of the sensitivity of solar sensors is noticeable.

**Tab. 2. Basic technical description of the sensors. Measurement owner: "OU" means University of Ostrava, "PHI" means Public Health Institute Ostrava, "CM" means custom made sensors.**

| Measured quantity | Bandwidth [nm] | Unit | Type | Company | Measurement owner |
|---|---|---|---|---|---|
| Broadband irradiance | 400–1100 | W m$^{-2}$ | EMS 11 | EMS Brno, CR | OU |
| UVB | 280–315 | W m$^{-2}$ | SKU 430 | Skye, UK | OU |
| UVA | 315–380 | W m$^{-2}$ | SKU 420 | Skye, UK | OU |
| PAR | 400–700 | μmol m$^{-2}$ s$^{-1}$ | EMS 12 | EMS Brno, CR | OU |
| 510–700 nm | 510–700 | μmol m$^{-2}$ s$^{-1}$ | CM | EMS Brno, CR | OU |
| 600–700 nm | 600–700 | μmol m$^{-2}$ s$^{-1}$ | CM | EMS Brno, CR | OU |




| Red (660 nm) | 610–680 | µmol m$^{-2}$ s$^{-1}$ | SKR 110 | Skye, UK | OU |
|---|---|---|---|---|---|
| Far red (730 nm) | 690–760 | µmol m$^{-2}$ s$^{-1}$ | SKR 110 | Skye, UK | OU |
| Air temperature | - | °C | EMS 33R | EMS Brno, CR | OU |
| Relative air humidity | - | % | EMS 33R | EMS Brno, CR | OU |
| Air temperature | - | °C | resistive sensor | Thies Clima, Germany | PHI |
| Relative air humidity | - | % | hygrometer | Thies Clima, Germany | PHI |
| Air pressure | - | hPa | anaeroid | Thies Clima, Germany | PHI |
| Wind speed and direction | - | m s$^{-1}$; ° | Windsonic | Gill Instruments, UK | PHI |
| PM$_{10}$ | - | µg m$^{-3}$ | 5030 | Sharp, Japan | PHI |
| NO$_x$ | - | µg m$^{-3}$ | APNA 370 | Horiba, Japan | PHI |
| SO$_2$ | - | µg m$^{-3}$ | APSA 370 | Horiba, Japan | PHI |

Additional meteorological variables, such as air temperature, relative air humidity, air pressure, wind speed and wind direction and concentrations of PM$_{10}$ (dust particles with size greater than 10 µm), NO$_x$ (nitrogen oxides) and SO$_2$ (sulphur dioxide), were measured by the Public Health Institute Ostrava (Tab. 2). The nearest measuring station (49.819031°N,

18.340355°E) was located approximately 1.7 km from BG OU. Concentrations of air pollutants and meteorological variables were provided by Public Health Institute Ostrava as 1 h averages. The measurements were part of a routine process for monitoring air quality in Ostrava. In this routine, series of measurements are checked twice per day, each station is physically checked three times per week, the check and calibration of all sensors is performed by certified companies once per year, and the calibration of sensors in the calibration laboratory takes place once every two years. Measurements are

controlled in three different procedures that occur *i)* daily, *ii)* once per month before sending to air pollutants database ISKO, and *iii)* once per year. For the sake of simplicity in format of each record, the 1 h measurements were repeated for all six 10 min records falling in this hour. There was no measurement provided by Public Health Institute Ostrava near the CHMI area in Poruba and air quality data measured by CHMI was not available for free usage. This is the reason why the database does not contain additional meteorological variables for the location in Poruba.

Three additional pieces of information are included in the database. The first piece is the season when the measurement was done: spring for the term 21$^{st}$ March–20$^{th}$ June, summer for the term 21$^{st}$ June–22$^{nd}$ September, autumn for the term 23$^{rd}$ September–20$^{th}$ December, and winter for the term 21$^{st}$ December–20$^{th}$ March. The second piece is the prevailing weather conditions for each record with three categories: cloudy, partly cloudy and sunny. It was determined by the analysis of the daily profile of the broadband irradiance. Fig. 2 exhibits daily profiles for three different days in March 2015; sunny days

offer a bell-shaped profile, cloudy days have flat low profiles and partly cloudy exhibit large variability within the day. The third piece is the air pollution category and it is available only for the stations S1 and S2. It was established on the basis of intervals of concentration of PM$_{10}$ in the air at the measuring station of Public Health Institute Ostrava according to Elshout et al. (2008). There are five categories of air pollution: very low (0-11 µg m$^{-3}$), low (12-24 µg m$^{-3}$), medium (25-49 µg m$^{-3}$), high (50-99 µg m$^{-3}$) and very high (greater than 100 µg m$^{-3}$).




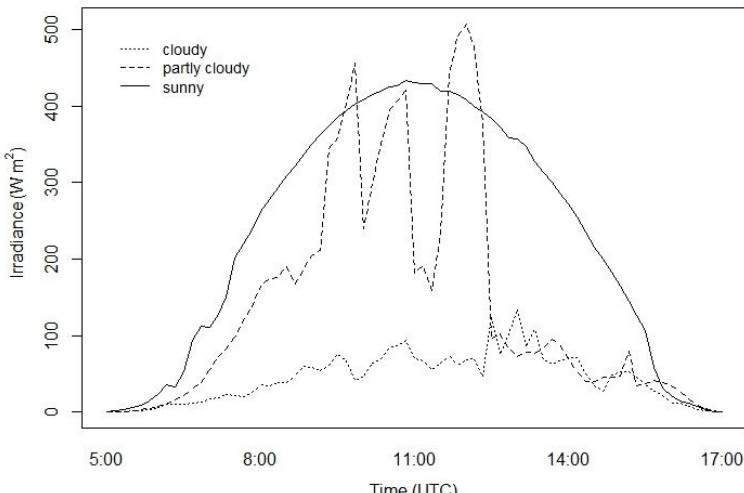

**Fig. 2. Example of daily profiles of the broadband irradiance for three different days in March 2015 (sunny, cloudy and partly cloudy days).**

## 3 Quality check applied to radiation data

To the best of the knowledge of the authors, there has not been published and accepted procedure for checking the quality of irradiances in the spectral bands of interest yet. In the absence of such a procedure, we have adapted the approach of Korany et al. (2016) which applies to measurements of global and diffuse total irradiances. In our specific case, the quality check

(QC) of Korany et al. consists in checking that each measurement $E(t, \Delta\lambda)$ falls into the range [0.03 $E0(t, \Delta\lambda)$, $E0(t, \Delta\lambda)$], where $t$ is the time of measurement, $\lambda$ the wavelength, $\Delta\lambda$ the spectral range, $E(t, \Delta\lambda)$ the measurement, and $E0(t, \Delta\lambda)$ the corresponding irradiance or PPFD at the top of the atmosphere (TOA). Given a typical spectrum $E0(\lambda)$ (Kurucz, 1992) of the irradiance at TOA, the ratio of the irradiance at TOA for a given spectral band $E0(\Delta\lambda)$ to the total irradiance at TOA $E0$ was computed for each spectral band (Tab. 3). It was checked that these ratios are fairly constant by performing similar

calculations with other reference spectra of TOA irradiance found in libRadtran package (Mayer & Kylling, 2005). In addition, the Kurucz spectrum was converted into spectral PPFD in µmol m$^{-2}$ s$^{-1}$ nm using the equation $E0(\lambda) = h\ c/\lambda$, where $E0(\lambda)$ is the spectral irradiance, $h$ is the Planck constant and $c$ is the velocity of the light. The PPFD $E0(t, \Delta\lambda)$ in µmol m$^{-2}$ s$^{-1}$ was obtained when needed by summing up the spectral PPFD within the band (Tab. 3). The coefficient for the unit conversion for each spectral band was obtained by ratioing the irradiances expressed in both units (Tab. 3). Finally, the final

coefficient for converting $E0$ into $E0(\Delta\lambda)$ was computed by multiplying the ratio to the total irradiance by the coefficient for the unit conversion (Tab. 3). Time series of the total irradiance at TOA in a 1 min step were downloaded from the website



soda-pro.com (http://www.soda-pro.com/web-services/radiation/cams-radiation-service) for both locations and then averaged every 10 min to match the measurements. By applying the appropriate coefficient for each spectral range, corresponding time-series of $E0(t, \Delta\lambda)$ were obtained against which the measurements $E(t, \Delta\lambda)$ may be compared for the first step of the QC.

The second step was performed aiming at flagging measurements that are significantly different from 0, i.e. measurements that are greater than 1.5 times the measurements uncertainty given by the World Meteorological Organization (WMO, 2012). As this uncertainty is based on 95% probability, multiplying it by 1.5 and assuming a Normal law for uncertainty means that there is a 99.7% chance that a measurement greater than this threshold is significantly different from 0. The WMO guide provides relative uncertainty in percent for hourly values of total irradiation. We have estimated the threshold for each
spectral band as follows. The relative uncertainty for daily irradiation of good quality is set to 5% in the WMO guide if the irradiation is greater than 8 MJ m$^{-2}$, which corresponds to an hourly mean of irradiance of 220 W m$^{-2}$ for an average day length. The 95% relative uncertainty for hourly values in the WMO guide is 8%, i.e. approximately 18 W m$^{-2}$. Multiplying by 1.5 yielded the threshold for the total irradiance: 27 W m$^{-2}$ reported in Tab. 3. In a similar way to step 1, the threshold for each spectral range was computed by multiplying the threshold for the total irradiance by the final coefficient, see Tab. 3.

**Tab. 3. Irradiance for each spectral band at the top of the atmosphere (TOA, in W m$^{-2}$), its ratio to the total irradiance, conversion coefficients from W m$^{-2}$ to µmol m$^{-2}$ s$^{-1}$ when needed, final coefficients for calculation of irradiance in spectral bands, and resulting**
**upper thresholds.**

| Spectral band | TOA Irradiance W m$^{-2}$ | TOA Irradiance µmol m$^{-2}$ s$^{-1}$ | Ratio to total irradiance | Unit conversion coefficient | Final coefficient, including unit conversion | Thresholds |
|---|---|---|---|---|---|---|
| Total irradiance | 1365.780 | - | 1 | - | 1 | 27 W m$^{-2}$ |
| Broadband 400-1100 nm | 908.205 | - | 0.665 | - | 0.665 | 18 W m$^{-2}$ |
| UVB | 20.386 | - | 0.0149 | - | 0.0149 | 0.4 W m$^{-2}$ |
| UVA | 65.355 | - | 0.0479 | - | 0.0479 | 1.3 W m$^{-2}$ |
| 400-700 nm | 534.311 | 2432.153 | 0.39121 | 4.552 | 1.7808 | 48 µmol m$^{-2}$ s$^{-1}$ |
| 510-700 nm | 327.785 | 1644.228 | 0.24000 | 5.016 | 1.2039 | 33 µmol m$^{-2}$ s$^{-1}$ |
| 600-700 nm | 160.760 | 870.742 | 0.11771 | 5.416 | 0.6375 | 17 µmol m$^{-2}$ s$^{-1}$ |
| 660 nm | 131.406 | 701.406 | 0.09621 | 5.338 | 0.5136 | 14 µmol m$^{-2}$ s$^{-1}$ |
| 730 nm | 95.673 | 578.752 | 0.07005 | 6.049 | 0.4238 | 11 µmol m$^{-2}$ s$^{-1}$ |



The QC for air temperature and relative air humidity was done by checking if data belongs to the interval of values which were trustable for the measuring period at measuring places (nonsense values were eliminated). The QC for additional meteorological data and air quality measurements has been described in Sect. 2.2.

# 4 Results

## 4.1 Results of the quality check

There were a total of 131 142 records for station S1, 131 558 records for station S2, and 130 659 records for station S3 from July 2014 to December 2016. Percentage of data greater than thresholds is greater than 30 %, except UVB (second column,
Tab. 4). This effect is explained by the fact that the records also include the night measurements which were close to zero.
High percentages are observed in broadband, PAR, and UVA. The lowest percentage of data greater than threshold value is observed in UVB region –17.8%. This fact was likely due to a too high threshold value for this spectral region (0.4 W m$^{-2}$), but measurements of UVB radiation are known to be difficult (e.g. Sayre & Kligman, 1992).

Average percentage of data greater than the threshold and passing QC based on the interval [0.03*E0, E0] was 98.8% (Tab. 4). The data at S3 had less percentage of data which successfully passed QC than the other stations. Differences are less than
3%; there is no apparent explanation.

Tab. 4. Data success rate for the QC. Rates are expressed as percentage. *TR* means threshold values from Table 3. The column
>*TR* gives the percentage of data greater than TR calculated from all records. *iQC* means the interval [0.03*E0, E0]. The column
>*TR* + <*iQC* gives the percentage of data <0.03 E0 calculated from numbers of data >TR. The column >*TR* + *iQC* gives the
percentage of data belonging to iQC calculated from numbers of data >TR. The column >*TR* + >*iQC* gives the percentage of data
>E0 calculated from numbers of data >TR.

|  | >TR | >TR + <iQC | >TR + iQC | >TR + > iQC |
|---|---|---|---|---|
|  | % | % | % | % |
| **S1** |  |  |  |  |
| Broadband | 43.32 | 0.03 | 97.79 | 2.19 |
| PAR | 41.88 | 0.03 | 99.97 | 0.01 |
| 510–700 nm | 41.48 | 0.03 | 99.97 | 0.01 |
| 600–700 nm | 40.83 | 0.03 | 99.97 | 0.00 |
| 660 nm | 28.95 | 0.59 | 99.42 | 0.00 |
| 730 nm | 30.25 | 0.48 | 99.52 | 0.00 |
| UVA | 38.63 | 0.06 | 99.95 | 0.00 |
| UVB | 17.31 | 1.38 | 98.62 | 0.00 |
| **S2** |  |  |  |  |



| | | | | |
|---|---|---|---|---|
| Broadband | 43.51 | 0.02 | 97.01 | 2.97 |
| PAR | 41.83 | 0.03 | 99.97 | 0.00 |
| 510–700 nm | 41.81 | 0.03 | 99.96 | 0.01 |
| 600–700 nm | 41.21 | 0.03 | 99.96 | 0.01 |
| 660 nm | 28.77 | 0.59 | 99.41 | 0.00 |
| 730 nm | 29.92 | 0.44 | 99.56 | 0.00 |
| UVA | 38.92 | 0.06 | 99.95 | 0.00 |
| UVB | 17.84 | 1.30 | 98.68 | 0.02 |
| S3 | | | | |
| Broadband | 44.02 | 0.00 | 94.74 | 5.25 |
| PAR | 42.53 | 0.01 | 99.76 | 0.24 |
| 510–700 nm | 42.37 | 0.01 | 99.73 | 0.27 |
| 600–700 nm | 41.92 | 0.03 | 99.70 | 0.27 |
| 660 nm | 31.22 | 0.71 | 99.29 | 0.00 |
| 730 nm | 32.22 | 0.62 | 99.40 | 0.00 |
| UVA | 38.74 | 0.08 | 99.92 | 0.00 |
| UVB | 18.40 | 1.34 | 93.22 | 5.45 |

## 4.2 Additional comments

Some values are greater than the corresponding irradiance at TOA. These values were usually connected with partly cloudy weather, when multiple scattering on the cloud edges could increase incident solar irradiation (Mims & Frederick, 1994).

Because of noticeable shading effects experienced by the three stations, the direct component of the radiation is extinguished when the sun is below the local horizon. In addition, a part of the sky is always shaded, so a part of diffuse radiation is always missing in the measurements, whose magnitude depends on the day and time of the day and atmospheric constituents. Moreover, these objects reflect the downward radiation and part of the reflected radiation may contribute to the measured

10    radiation. This effect can potentially increase the measured radiation. It is very difficult to quantify it as it depends on optical properties of the reflecting object, day and time of the day and atmospheric constituents. Therefore, our measured data is influenced by these effects and it can be used only for purposes where these effects may be neglected.



## 4.3 The database: description and where to access it

One file containing 10 min averages of irradiance, meteorological variables and air quality observations was created for each of the three surface stations for each of the three years. These nine files can be downloaded from the Pangaea website using a unique identifier (https://doi.pangaea.de/10.1594/PANGAEA.879722). To avoid confusion, time stamps are given for both the beginning and the end of each 10 min averaging period. These files are self-explanatory and contain 40 columns for station S1, 39 columns for station S2 and 29 columns for the station S3 with these values: code for station, date/time (in UTC), date/time start and end, year, solar azimuth angle, solar elevation angle, mask shadow (which gives information if measured data were influenced by shade of surrounding trees – labelled by "0", or not – labelled by "1"), season (spring, summer, autumn, winter), weather category (cloudy, partly cloudy, sunny; determined according to a daily course of global radiation), air pollution category (very low, low, medium, high very high; determined according to the concentration of $PM_{10}$), relative air humidity (in %), air temperature (in °C), precipitation (in mm), broadband irradiance ([400, 1100] nm; in W m$^{-2}$), PPFD in the spectral bands [400, 700] nm (PAR; in µmol m$^{-2}$ s$^{-1}$), [510, 700] nm (in µmol m$^{-2}$ s$^{-1}$), [600, 700] nm (in µmol m$^{-2}$ s$^{-1}$), [610, 680] nm ( 660 nm; µmol m$^{-2}$ s$^{-1}$), [690, 760] nm ( 730 nm; µmol m$^{-2}$ s$^{-1}$), irradiance in UVA region ([315, 380] nm; in W m$^{-2}$), and in UVB region ([280, 315] nm; in W m$^{-2}$). Each radiation measurement has one column with a quality flag, which gives information if measured value of spectral irradiance passed quality check. It means if it falls into the interval [0.03 E0, E0], where E0 is the corresponding spectral irradiance at TOA, and at the same time if it was greater than the minimum: "1" means that the measurement falls in the interval, and "0" means that it is outside. All data including night measurements is present in the datasets. Figures with the influence of the horizon on the measuring stations are included in each dataset. Dataset for the station S1 contains additional columns with meteorological data measured by Public Health Institute Ostrava: concentrations of $PM_{10}$, $SO_2$, $NO_x$, NO, $NO_2$ (in µg m$^{-3}$), air temperature (in °C), relative air humidity (in %), air pressure (in hPa), wind speed (in m s$^{-1}$), and wind direction (in °). Data measured by Public Health Institute Ostrava had a native 1 h resolution, so it was copied to all six 10 min records of data measured by the University of Ostrava for each hour. This data can be connected with the dataset for station S2 because of identical location. For station S3, this data was not available from Public Health Institute Ostrava.

## 5 Summary and perspectives

A homogeneous database of 10 min measurements of downward solar irradiance in several ultraviolet and visible bands has been assembled. The data covers the period from July 2014 to December 2016 and originates from three ground stations representing different air pollution conditions in the Ostrava region (Czech Republic). This quality-checked data is available online at https://doi.pangaea.de/10.1594/PANGAEA.879722. The data provides an insight to a spatial and temporal variability of solar radiation in spectral bands [280, 315] nm (UVB), [315, 380] nm (UVA), [400, 700] nm PAR), [510,

700] nm, [600, 700] nm, [610, 680] nm (660 nm), [690, 760] nm (730 nm), and [400, 1100] nm (broadband). Because of the specific environmental conditions of the stations, this data has some limitations that should be taken into account before use. Because of their high temporal resolution and the period spanning more than two years, this data constitutes a precious tool for the estimation of the radiation environment in an industrial city in the middle of Europe and it can be used as input data for models of influence of this radiation regime on plants. One of the goals could be to find the influence of atmospheric pollution on the spectral composition of incident solar radiation with a focus on analyses of differences between measured values. It is possible to calculate the PPFD in blue [400, 510] nm: the PAR PPFD minus the PPFD in [510, 700] nm) or in green [510, 600] nm: the PPFD in [510, 700] nm minus the PPFD in [600, 700] nm). Hence, changes and differences in irradiances and PPFDs in these spectral bands can be studied in different environment conditions. The understanding of the radiation environment is crucial in relation to vegetation and its ecological services in cities which are important for a correct function of the microclimate of urban areas. In the near future, analyses of selected spectral ratios in relation to atmosphere pollution are planned. The measurements of solar irradiation will continue and new data can be added to this database later, thus longer and detailed studies of time series will become available for this location of an industrial city in the middle of Europe.

## Acknowledgement

This work was financially supported by COST CZ LD14005, Institutional Research Support grants (SGS06/UVAFM/2016 and SGS05/UVAFM/2017) from the University of Ostrava, by EU structural funding Operational Programme Research and Development for Innovation, project no. CZ.1.05/2.1.00/19.0388, and by the Ministry of Education, Youth and Sports of the Czech Republic in the "National Feasibility Program I", project LO1208 "TEWEP". We thank to Public Health Institute Ostrava and EMS Brno for the cooperation. We thank to Amelie Driemel for her valuable comments. We thank to Vladimír Bradáč for his English language assistance. We thank the anonymous reviewers for their helpful comments and suggestions.

## Author contributions

L. Wald suggested possibilities of data quality check. P. Blanc determined a shading effect on measuring stations according to combinations of solar zenith and elevation angles and horizon. L. Wald, P. Blanc, M. Navrátil and V. Špunda participated in the manuscript preparation. M. Opálková did the operation and maintenance of the sensors and designed the data set, made a data quality check and prepared the manuscript with contributions from all co-authors.

## Competing interests

The authors declare that they have no conflict of interest.



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
