# Peer review of "A Database of 10 min Average Measurements of Solar Radiation and Meteorological Variables in Ostrava, Czech Republic"

_Earth System Science Data, 2017_

## Referee Comment (RC1) · Anonymous Referee #1 · 22 Jan 2018

Review of paper ÂńÂăA database of 10mn average measurements . . ., Czech Republic ÂăÂż by Opalkova et al

General comments This paper presents a newly set-up database of solar radiation plus few meterological and pollutants measurements carried in Ostrava, NE of the Czech Republic. This database has been built by scientists from the University of Ostrava in collaboration with french scientists which expertise in the area of solar radiation measurements and analysis is acknowledged. The sites, sensors used and their maintenance, and the quality checks performed on the raw data and their results are detailed. In particular authors propose a new procedure for checking the quality of irradiances

in specific spectral bands which is an adaptation of procedures used for QC of total irradiances.

Specific comments As I'm not an expert in solar radiation data QC I can not fully judge the validity of the new procedure proposed. I therefore mainly comment on other aspects as rationale, statistics, and form

RationaleÂă: The stress is put on the utility of the measurements set-up as a mean to understand the impact of variations in solar radiation, especially those related to atmospheric pollutants, on vegetation in Ostrava. Have the authors any idea of the surface of the city, presented as industrial, occupied by vegetation and if this is less, in the average or more than similar citiesÂă? Rather than impacting the vegetation in the city itself, don't the authors think that given air-mass movements, the neighbouring areas ie the gardening belt of the city might be impacted and your measurements if representative or extrapolable would be useful to assess also crops sensitivity to variations in PAR due to pollution. Lastly what might be the counter-effect of pollutants deposits vs increase in diffuse radiation on vegetation photosynthetic capacityÂă? Similarly with regards to health issues what might be the balance between a lower exposure to UVA/UVB due to attenuation by pollution vs breathing these aerosols ... I would like the authors to expand a bit their introduction considering these aspects to enhance the rationale of their in-situ measurements.

StatisticsÂă: At the beginning of section 2.1, authors give mean values of sunshine duration air T° ... first we don't understand if this is for Ostrava or the Czech republic. Second it would have been interesting to provide also these values as obtained from you in-situ measurements even if 3 years are available only. In section 2.2 authors present additional pieces of information provided in the database, in particular the ÂńÂătype of weatherÂăÂż. Yet they do not explain how they have assigned each day to one of the three type of weather. What are the statistics and distance metrics usedÂă? Supervised classificationÂă? Amplitude and variance of the diurnal cyclesÂă? This must be developed and explained. For the information relative to the season I do not

really understand the usefullness of it. Can you explain it a bitÂă?

MaintenanceÂă: In section 2.2 authors also describe the way sensors are maintained and the frequency of these maintenances (ech one or two month depending on the season). From the QC performed can you infer that this frequency is high enoughÂă? How far the measurements rejected by the QC correspond to hours when the maintenance was operatedÂă? Have the authors the exact dates / hours of maintenanceÂă? If yes these dates / hours should be reported in the database in a dedicated column so that users can exactly know if the erroneous data (i.e those not passing the QC) are due to maintenance (and maybe do interpolations from the hours preceding / following the maintenance) or an another failure.

Technical corrections

AbstractÂă: lines 14-17Âă: the authors should better stress there that they propose a new procedure for QC of irradiance in different spectral bands (cf beginning of section 3).

IntroductionÂă: page 2 last sentenceÂă: This sentence is confusingÂă: we don't understand if you propose to extrapolate the measurements or the procedure to other regions and which ones exactly (what are the ÂńÂăregions similarÂăÂż to Ostrava Âă?)Âă?. please reword.

Measurements sitesÂă: on the whole I find very difficult to follow / understand what are the sites you speak about along the whole paper. The way you name and call them is confusing. First starting from the map in figure 1 I would label the two sites ÂńÂăBGOU (S1, S2)ÂăÂż and ÂńÂăCHMI (S3)ÂăÂż than 1 and 2. This would be very very helpfull. + add the coordinates on the map. Without these coordinates I can not figure out where is the PHI site ( please if possible locate it on the map in figure 1 as well) which is important with regards to atmospheric dynamics and dominant winds ... Pictures of the sites – unless confidential - would be a nice supplementary material to provide on the PANGEA website

For easing the review please do not present a same table on two different pages (a bit annoying) and number the lines with a step of two.

P3 : lines 26-27 : I couldn't find any info about altitude in Tab1 or Fig 1 so delete Âń (Tab1, Fig1) Âż.

P4, line 13 (and elsewhere in the paper)Âă: please be more precise about datesÂă: from the 1st of July 2104 + hour to the 31st of December 2016 + hour (since in some databases records are provided for non complete years).

P5, Line 17Âă: as the ÂńÂă2ÂăÂż stations were only 3m apart

P6, Line 4Âă: Within its network of X (? please provide this number) stations, the station the nearest of BGOU is located approximately at 1.7km (GPS coordinates)

P6, Line 11Âă: For the sake of simplicity → remove this sentence which is unnecessary here (explanations provided later in the paper)

P6, lines 13 and 14 Âă: change ÂńÂăCHMI area in PorubaÂăÂż for ÂńÂăCHMI stationÂăÂż and change ÂńÂăthe location in PorubaÂăÂż for ÂńÂăCHMIÂăÂż.

P6, line 19Âă: broadband irradiance as exemplified in Fig.2 which presents profiles . . .

P7, line 15Âă: LibRadtran software (and not a package of software as R or Matlab . . .)

P8, line 1Âă: for ÂńÂăBGOU and CHMI sitesÂăÂż (instead of both locations)

P9, line 7Âă: add ÂńÂă(BGOU) ÂńÂăafter S2 and ÂńÂă(CHMI)ÂăÂż after S3 line 13Âă: change for ÂńÂăthe station at CHMI (S3) had . . . than ÂńÂăstations at BGOU (S1, S2)ÂăÂż

P10, last sentenceÂă: use the plural form. Can these effects be neglected for your study purposes i.e. impact of SR variability on vegetationÂă?

P11Âă: lines 6 – 15Âă: move that in a table lines 16-18Âă: delete ÂńÂăIt means if it . . . minimumÂż. explanations given before in the paper. Line 19Âă: ÂńÂăareÂăÂż

present + I couldn't find the figures in the files I uploaded. You should rather provide them as supplementary files just as the figure done from google earth which present the shading effect Line 23-25Âă: Data ÂńÂăof air pollutants and meteorological parametersÂăÂż measured by …. ÂńÂătheseÂăÂż data.

P12, line 5Âă:for ÂńÂămodeling theÂăÂż influence line 9Âă: studied in different environment conditionsÂă: please be more preciseÂă: meteorological and air pollution conditions I guess … line 10Âă: reword this sentence I don't understand what is a ÂńÂăcorrect function of microclimateÂăÂż line 11Âă: spectral ratiosÂă? Dio you really mean ratios or bandsÂă? If you mean ratio please give example of bands you could use to compute ratios …

Tab 2Âă: wonder if you should not split the table into two because it is confusing with regards to the sites where the instrumentys are implemented. For what I understand all instruments belonging to OU are on sites S1,S2 and S3 whereas instruments belonging to PHI are on a site 1.7km from BGOU

Tab 4Âă: legendÂă: numbering your columns would ease the reading of the table. You should also add BGOU and CHMI after S1/S2 and S3.

Fig2. LegendÂă: please provide the dates of these three days of March 2015 ….

---

## Referee Comment (RC2) · Anonymous Referee #2 · 29 Jan 2018

General comments

The paper aim was to provide complex information on solar radiation, air pollution and meteorological data measured in Ostrava and presented as free dataset in the PANGEA database for any user. The main advantage of the paper is establishment of solar radiation measurements in different spectral bands. The data can be used to study relationships between them in industrial polluted area. The data set provides good platform for further measurements and modelling. High attention was payed to Quality control methods used for good data selection. Detailed and relevant information on the measurements performance, data processing and control for possible data

users is very important. However, in my opinion, it was not fulfilled completely and correctly. I suggest several major (Specific comments 1) and minor corrections (Specific comments 2) of the paper and after that next revision.

Specific comments 1:

1. Database purpose P. 1 abstract: This database offers a unique ensemble of variables having a high temporal resolution and it is a reliable source of information on radiation in relation with environment and vegetation in highly polluted areas of industrial cities in the middle of Europe. P. 11: it can be used as input data for models of influence of this radiation regime on plants

Please, explain how this database can be used for research targeted to study of influence of polluted urban environment on plant when there have not been presented biological measurements or observations? In my opinion, the data from presented period can be used to study relationships between measured radiative parameters in polluted area of north middle latitudes under different condition (solar zenith angle, wind condition, relative humidity etc.). The data can be used for any environmental modelling, e.g. for atmospheric chemistry models, urbanistic studies - not only for biological research. In the Introduction, there were presented many works studying relations between biological processes and selected ratios UVB/PAR, UVA/PAR or DIF/GLO. I recommend presentations of some relations between irradiances or photon fluxes in different spectral bands in this paper to attract the database users.

Please, specify the aim of this database creation and possibilities of the data utilisation in abstract and introduction. With respect to the database purpose, present relevant references in the Introduction (you referred only biological research).

2. Missing proof about pollution differences between presented stations

P. 12: One of the goals could be to find the influence of atmospheric pollution on the spectral composition of incident solar radiation with a focus on analyses of differences

between measured values. P. 3: BG OU is situated approximately 3 km from an industrial area which produces many air pollutants (Jančík et al., 2013) and is much more influenced by air pollution than the CHMI location, especially in the winter months.

There are mentioned 3 localities where data was measured – 2 stations in the Botanical garden of the OU and third about 3 km far on the CHMI plot. Stations are very close. There is declared that S3 station is in less polluted area than S1 and S2 stations. It is necessary to give some proof about this conclusion (some analysis of differences btw. stations) and present it in the paper. The CHMI air quality monitoring network data can be used for this purpose. Why there were established 2 stations in the Botanical garden so close each to other? Explain it in the paper. To study influence of air pollution on solar radiation spectral distribution, at least one station should be placed in rural unpolluted area with similar geographical characteristics as at stations in Ostrava. Some air pollution indicator, especially aerosol content, should be measured at every station. Air pollution characteristics were measured at fourth station (within very small area studied, these data do not represent neither botanical garden stations S1 and S2 nor the CHMI station S3) and it should be clearly explained. I suggest introduction of this station S4 characteristics in the explanatory tables 1 and 2 and in the map in Fig.1. If S1-S3 stations represent similar pollution condition (with characteristics measured at the S4) then reflect it in abstract and text ( see also point 1).

3. Unification of data description in the paper text, tab. 2 and in the database

In table 2 there is 'broadband irradiance' but in database 'shortwave downward global irradiance' Red, blue, green band terms used in text, only red in table 2, UVA, UVB in text - UV-b, UV-a in the database etc. I recommend usage of the same terms for measured irradiances and photon fluxes in database, text and tables.

4. The threshold as QC control criterion

P. 7: The relative uncertainty for daily irradiation of good quality is set to 5% in the WMO guide if the irradiation is greater than 8 MJ m-2 , which corresponds to an hourly mean

of irradiance of 220 W m-2 for an average day length. Explain, please, term 'average day length' and how it was calculated/derived and for which geographical coordinates. Explain clearly what is the difference between data above and under the threshold you defined. You based your criteria for threshold on recommended but not real characteristics of your measurements. I disagree with the thresholds definition. If the widened uncertainty of measurements have been the concept for it, then uncertainties of every instrument provided by manufacturer or calibration authority should have been used (not the WMO data quality categorization). I suggest different threshold definition and its calculations performing separately for every measured radiative parameter (In that case 80% or realistic UVB data would not be under threshold limit.) and with reasonable explanation of the meaning of the criteria for data separation to above and under defined threshold values. If it would be impossible, I suggest exclusion of threshold concept from QC control. Why didn't you base the threshold calculations on the noise values of particular instrument?

5. Relative spectral response of sensors missing

Please, present the relative spectral responses of particular radiation sensors (don't let reader searching general information by internet). I recommend presentation in separate table (e.g in Appendix) together with information about source of this information (whether it was measured by manufacturer or calibration authority or presented by manufacturer as approximate characteristic of the instrument type). Other important characteristics of the sensors can be also added – time response, cosine errors etc.

6. Data complexity indicator missing

The radiative data were sampled every 1 min. This sampling interval is far from the WMO recommendation (1 s) and a lot of information about radiation variability was lost. 10 min averages are presented in the database. There is no indicator of data complexity. I suggest presentation of number of 1 min data involved for 10 min average calculation.
7. Offset presentation missing

I suggest presentation of night values from all sensors in the database which will help to quantify noise - influence of infrared radiation and data acquisition system on measured data.

8. Cloudiness condition in night hours

How did you characterise cloudiness condition in the night hours when there was no solar irradiance approaching sensors? Add some explanation to the paper.

Specific and technical comments 2:

1. P. 1 abstract: '10 min of downward surface irradiance ', revise this term with respect to points 3 and 4. Where sensors placed on the surface? 2. P.1 abstract: These two stations offer additional data: PM10, SO2, NOx, NO, NO2 concentrations. – revise the sentence with respect to point 2 in the previous part of revision - air pollution data were measured at 4th station. 3. P. 4: The PPFDs in three PAR bands were calculated from the sensor data: blue [400, 510] nm, green [510, 600] nm, and red [600, 700] nm by subtraction. Which radiation characteristics were obtained by subtraction of values measured in some spectral bands? It seems that all parameters in Tab.2 were measured. Please, explain the meaning of the sentence. 4. It would be valuable to have photos of instrument installation at particular stations. 5. State the altitude of sensors above surface. 6. Please present the station (including coordinates) where the long-term climate characteristics came from (part 2.1). 7. There is mentioned that some obstacles reduced direct component of solar radiation (p. 10) . I recommend showing the horizon elevation as function of azimuth for stations with solar radiation measurement in this paper. I also recommend calculation of sun elevation and azimuth for every data, comparison with horizon altitude by particular azimuthal angles and evaluation of the shading indicator. 8. An altitude should be added to geographical characteristics. Solar radiation undergoes changes with altitude and it could be reason for differences in radiation measured at particular stations. 9. I recommend

presentation of typical wind condition at every station. Wind plays important role in aerosol and pollutant spreading. 10. In part 4.1, there is declared precipitation data storage in the database (and the data are there). In previous parts and tab. 2, there is no information how and where it was measured. Information on snow presence on the surface (or albedo data) would be valuable as auxiliary meteorological parameter because reflected irradiance contributes to diffuse component of measured global radiation significantly. If this information was available (at least at the CHMI S3 station), add it in the database. 11. P.7: Which ET spectrum was finally used - Kurutz (1992) or Mayer and Kylling (2005)? Which was the Sun –Earth distance when the spectra were measured? Is there difference in wavelength resolution in mentioned spectra? Was the integral ETC presented in table 3 obtained by integration of spectral data from Kurutz (1992) or Mayer and Kylling (2005) (if not, present the source of the value)? Explain calculation of integral ET irradiances and photon fluxes in selected spectral ranges in more details. 12. P. 5:..sensors measuring radiation in the intervals [510, 700] nm and [600, 700] nm contain cut-off filters which have the S-shaped permeability curve and it causes a little bit different measured values. Based on these tests, we can conclude that no long-term decrease of the sensitivity of solar sensors is noticeable.... Please explain the S shaped permeability curve relations to the filter and the sensitivity tests in the paper. Permeability is magnetic characteristics of materials. 13. P. 5: Each sensor was equipped with cosine correction – explanation necessary. 14. P. 6: Daily profile of global radiation... – Did you mean daily course? 15. P. 1: abstract: air temperature at the surface - Clarify the thermometer position - at altitude 2 m or more closely to the surface? 16. P. 5: No significant systematic biases were observed and the least square linear regression provided a cross-calibration correction of less than 5%. Differences between them could be caused by unequal irradiance during partly cloudy days, or by technical properties of sensors. ' unequal irradiance' - Did you mean variable irradiance? Please, explain how and why did you perform cross-calibration correction. Did you perform some calibrations of the sensors during the presented period? Please, describe the calibration methods. 17. P. 5: The term 'weather conditions' should be

replaced by term 'cloudiness condition'.

Other notes:

1. P. 5: In addition, at BG OU, measurements made by similar sensors were compared to check the temporal consistency as the stations were only 3 m apart. Coefficients of determination were in the interval [0.94, 0.98], thus confirming the expected similarity in data between S1 and S2. No significant systematic biases were observed and the least square linear regression provided a cross-calibration correction of less than 5%. This comparison between measurements of the same instrument type installed at the same place would have been perfect to organize before the beginning of measurements at particular stations. 2. The WMO recommends more frequent maintenance and control of instruments on site than once per month or 2 months. Are the instruments equipped with some ventilation to avoid persistence of water vapour condensation products (dew, freezing) on sensors? Cleaning of the instruments to avoid dust coverage on the sensors is recommended to perform more frequently in the future. Also levelling and dessicant checking should be provided more frequently. 3. Sensitivity of sensors operating in UV range of spectrum has been sometimes changing very rapidly and more frequent calibrations (at least once per year) are recommended. 4. P. 5: To check the long-term stability of the sensors, measurements from each of them were compared with the measurements in the broadband range and linear regressions were computed in the whole measuring period and for each individual year.

I tis not good method for stability check because operational broadband radiation sensor sensitivity can be also changing. Regular comparison to reference instrument is the WMO recommended procedure for the solar radiation sensor stability control.

Please also note the supplement to this comment:
https://www.earth-syst-sci-data-discuss.net/essd-2017-111/essd-2017-111-RC2-supplement.pdf

---

## Author Comment (AC1) · 23 Mar 2018

Dear Anonymous Referee,

we thank you for your evaluation of our manuscript and valuable comments. We have taken each of your comments and suggestions into account. We brought answers and changes to text to satisfy your concerns. The zip file is attached to this message and it contains the document with our responses to your concerns and comments (Response_Reviewer1_ESSD.pdf), the adjusted version of paper with highlighted changes in text (A database of 10 min average measurements of solar radiation and meteorological variables in Ostrava, Czech Republic_HIGHLIGHTED CHANGES.pdf), the ad-

justed version of paper in a clear form (A database of 10 min average measurements of solar radiation and meteorological variables in Ostrava, Czech Republic_CLEAR VERSION.pdf), and zip file "Appendix" with more documents (photos, pictures, graph, table).

Yours faithfully, Marie Opálková et al.

Please also note the supplement to this comment:
https://www.earth-syst-sci-data-discuss.net/essd-2017-111/essd-2017-111-AC1-supplement.zip